# Multifunctional CRISPR-Cas9 with engineered immunosilenced human T cell epitopes

Shayesteh R. Ferdosi [1,2,6], Radwa Ewaisha [1,3,6], Farzaneh Moghadam [4], Sri Krishna [1,4], Jin G. Park [1], Mo R. Ebrahimkhani [4,5], Samira Kiani [4] & Karen S. Anderson[1,3]

The CRISPR-Cas9 system has raised hopes for developing personalized gene therapies for complex diseases. Its application for genetic and epigenetic therapies in humans raises concerns over immunogenicity of the bacterially derived Cas9 protein. Here we detect antibodies to *Streptococcus pyogenes* Cas9 (SpCas9) in at least 5% of 143 healthy individuals. We also report pre-existing human CD8+T cell immunity in the majority of healthy individuals screened. We identify two immunodominant SpCas9 T cell epitopes for HLA-A*02:01 using an enhanced prediction algorithm that incorporates T cell receptor contact residue hydrophobicity and HLA binding and evaluated them by T cell assays using healthy donor PBMCs. In a proof-of-principle study, we demonstrate that Cas9 protein can be modified to eliminate immunodominant epitopes through targeted mutation while preserving its function and specificity. Our study highlights the problem of pre-existing immunity against CRISPR-associated nucleases and offers a potential solution to mitigate the T cell immune response.

[1] Center for Personalized Diagnostics, Biodesign Institute, Arizona State University, Tempe, AZ 85287, USA. [2] School of Molecular Sciences, Arizona State University, Tempe, AZ 85287, USA. [3] School of Life Sciences, Arizona State University, Tempe, AZ 85287, USA. [4] School of Biological and Health Systems Engineering, Arizona State University, Tempe, AZ 85287, USA. [5] Division of Gastroenterology and Hepatology, Mayo Clinic, Phoenix, AZ 85054, USA. [6]These authors contributed equally: Shayesteh R. Ferdosi, Radwa Ewaisha. Correspondence and requests for materials should be addressed to S.K. (email: samira.kiani@asu.edu) or to K.S.A. (email: Karen.Anderson.1@asu.edu)

The Clustered Regularly Interspaced Short Palindromic Repeat (CRISPR)-Cas9 technology has raised hopes for developing personalized gene therapies for complex diseases such as cancer as well as genetic disorders, and is currently entering clinical trials[1,2]. The history of gene therapy has included both impressive success stories and serious immunologic adverse events[3–8]. The expression of *Streptococcus pyogenes* Cas9 protein (SpCas9) in mice has evoked both cellular and humoral immune responses[9,10], which raises concerns regarding its safety and efficacy as a gene or epi-gene therapy in humans. These preclinical models and host immune reactions to other exogenous gene delivery systems[11–13] suggest that the pathogenic, non-self origin of Cas9 may be immunogenic in humans.

Both B cell and T cell host responses specific to either the transgene or the viral components of adenoviral[14,15] and adeno-associated viral (AAV)[11,12] vectors have been detected, despite relatively low immunogenicity of AAV vectors. In the case of AAV, specific neutralizing antibodies (Abs) and T cells are frequently detected in healthy donors[16–19] and specific CD8+T cells have been shown to expand following gene delivery[18]. There has been recent progress in developing strategies to overcome this problem, such as capsid engineering and transient immunosuppression[20–22]. The potential consequences of immune responses to expressed proteins from viral vectors or transgenes include neutralization of the gene product; destruction of the cells expressing it, leading to loss of therapeutic activity or tissue destruction; induction of immune memory that prevents re-administration; and fulminant innate inflammatory responses[23,24]. More potent immune responses to gene therapies have been observed in humans and non-human primate models compared to mice[8,25].

Of the Cas9 orthologs derived from bacterial species, the SpCas9 is the best characterized. *S. pyogenes* is a ubiquitous pathogen, with an annual incidence of 700 million worldwide[26], but the field is only now beginning to explore potential immunity to SpCas9 in humans[27,28]. CRISPR application for human therapies will span its use both for gene editing (through DNA double-strand breaks) or epigenetic therapies (without DNA double-strand breaks). In fact, recent reports shed light on CRISPR's ability to activate or repress gene expression in mice[29–31], which opens the door to a variety of new therapeutic applications such as activating silent genes, compensating for disrupted genes, cell fate reprogramming, or silencing disrupted genes, without the concern over permanent change in DNA sequence. However, unlike the use of Cas9 for gene editing, which may only require Cas9 presence in cells for a few hours, current techniques for CRISPR-based epigenetic therapies require longer term expression of Cas9 in vivo, possibly for weeks and months[30,31], which poses the challenge of combating pre-existing immune response towards Cas9. This challenge will need to be addressed before CRISPR application for human therapies, especially for epigenetic therapies, can be fully implemented. Delivery of CRISPR in vivo by incorporating its expression cassette in adeno-associated virus (AAV), will most likely shape many of the initial clinical trials as AAV-based gene delivery is one of the safest and most prevalent forms of gene therapies in human. AAV will enable longer term expression of Cas9, desirable for epigenetic therapies. Therefore, it is highly likely that CRISPR delivery through AAV and its expression within target cells will engage CD8+T cell immunity.

Here, we seek to characterize the pre-existing immune response to SpCas9 in healthy individuals and to identify the immunodominant T cell epitopes with the aim of developing SpCas9 proteins that have diminished capacity to invoke human adaptive response. We identify two immunodominant SpCas9 T cell epitopes for HLA-A*02:01 by an improved prediction algorithm and T cell assays using healthy donor PBMCs. We demonstrate that Cas9 protein can be modified to eliminate immunodominant epitopes through targeted mutation while preserving its function and specificity.

## Results

**Cas9-specific serum antibodies in healthy controls**. We first determined whether healthy individuals have detectable IgG Abs to SpCas9. Of 143 healthy control sera screened, 82 (57.3%) had detectable Abs against *S. pyogenes* lysate using ELISA (Fig. 1a). Sera with the highest reactivity to *S. pyogenes* lysate ($n = 80$) were screened for Abs against recombinant SpCas9, of which 7 (8.8%) were positive ($p < 0.0001$, two-tailed $t$-test). At least 5.0% of healthy individuals screened in this study had Cas9-specific Abs (Fig. 1a).

**Cas9 candidate T cell epitope prediction**. Whether Cas9-specific antibodies impact the efficacy or safety of CRISPR application in human remains to be seen. However, cellular immunity is expected to have a more significant impact in case of CRISPR delivery through viruses. The Cas9 expression cassette, delivered by a viral vector, leads to intracellular expression of this protein in target cells, which could evoke a cellular immune response. We thus focused on investigating the T cell immune response against SpCas9. We predicted HLA-A*02:01-restricted T cell epitopes derived from SpCas9 using a model that uses both MHC binding affinity and biochemical properties of immunogenicity[32] (Supplementary Table 1; the top 5 are shown in Fig. 1b). This model incorporates T cell receptor contact residue hydrophobicity and HLA binding prediction, which enhances the efficiency of epitope identification, as we previously reported[32]. We plotted the calculated normalized binding ($S_b$) and immunogenicity ($S_i$) scores for each peptide (Fig. 1c) to predict the more immunogenic epitopes, which are expected to have both high HLA binding (low $S_b$) and more hydrophobicity (high $S_i$). We chose HLA-A*02:01 because it is the most common HLA type in European/North American Caucasians.

**T cell epitope mapping of Cas9**. We then investigated whether peripheral blood mononuclear cells (PBMCs) derived from healthy individuals had measurable T cell reactivity against the predicted SpCas9 MHC class I epitopes. We synthesized 38 peptides (Supplementary Table 1) and grouped them into 10 pools of 3–4 peptides each. We measured peptide-specific T cell immunity using IFN-γ secretion ELISpot assays with PBMCs derived from 12 healthy individuals (HLA-A*02:01, $n = 10$; non-HLA-A*02:01, $n = 2$) and identified immunoreactive epitopes within pools 3 or 5 in 83.0% of the donors tested (90% of the HLA-A*02:01 donors; Fig. 1d). The seven individual peptides from pools 3 and 5 were evaluated by IFN-γ ELISpot and the dominant immunogenic epitopes were SpCas9_240–248 and SpCas9_615-623, designated peptides α and β, from pools 5 and 3, respectively. The subdominant epitopes were found to be γ and δ from pools 3 and 5, respectively. Both peptides α and β are located in the REC lobe of the Cas9 protein (Fig. 1e) that binds the sgRNA and the target DNA heteroduplex[33]. The individual peptides within pools that were positive for any donor were evaluated for this donor by IFN-γ ELISpot. The immunoreactivity and position of the 38 predicted peptides (a few of which are overlapping) within the Cas9 protein are shown in Fig. 1e.

Peptides α and β are shown as red dots on the epitope prediction plot (Fig. 1c) and their sequences and predicted ranking are shown in Fig. 1b and Supplementary Table 1. As predicted, these peptides had low $S_b$ and high $S_i$ values. Both the immunodominant (α and β) and subdominant (γ and δ) T cell

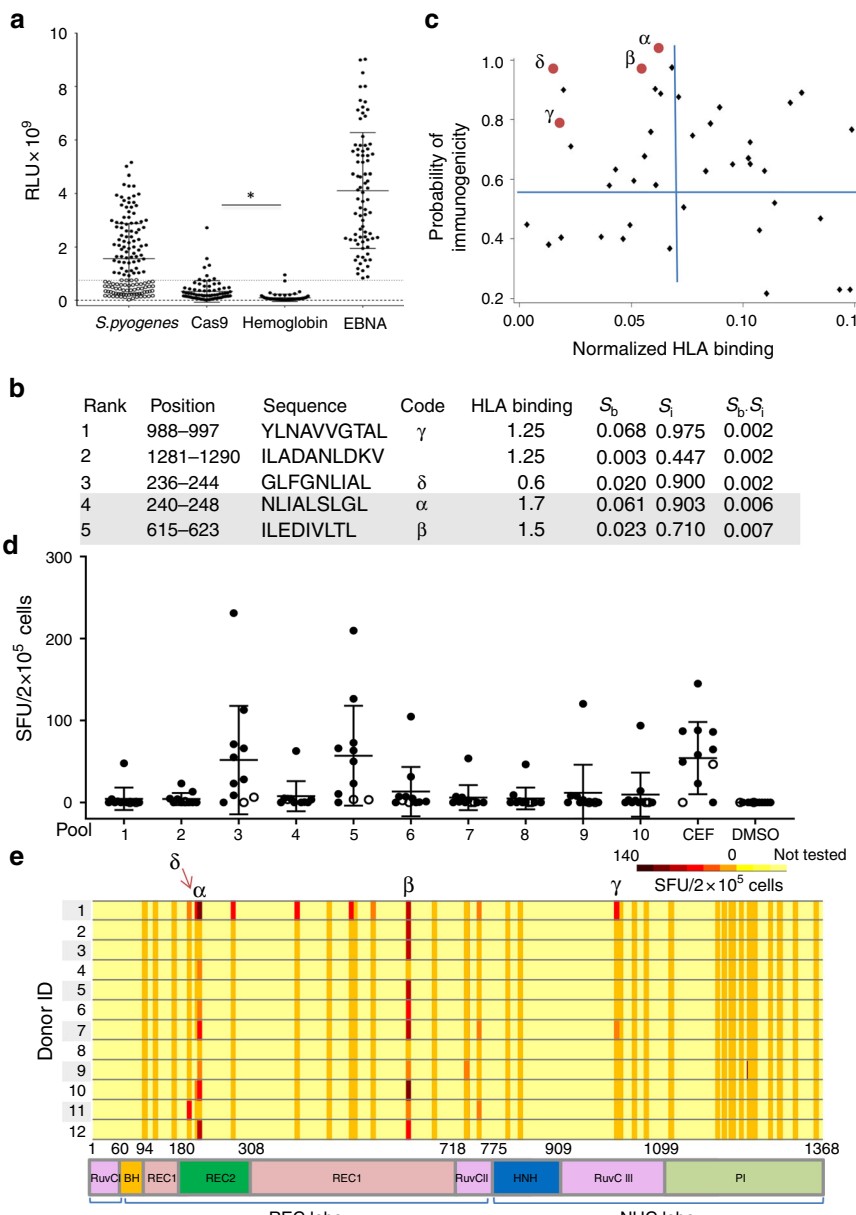

**Fig. 1** Detection of pre-existing B cell and T cell immune responses to SpCas9 in healthy donors and identification of two immunodominant T cell epitopes. **a** Specific serum Abs were detected against *S. pyogenes* lysate in 57.3% (*n* = 82) of 143 healthy controls. Sera with the highest reactivity to *S. pyogenes* lysate (*n* = 80, black circles) were screened for Abs against recombinant SpCas9, recombinant EBNA-1 protein (positive control), and human hemoglobin (negative control), of which 7 (8.8%) were positive for SpCas9 (above the dotted line; *\*p* < 0.0001). **b** The top 5 predicted SpCas9 T cell epitopes and their predicted $S_b$ and $S_i$ scores and ranking (based on the $S_b.S_i$ value)[32]. These top 5 peptides include the identified immunodominant (α and β; gray) and subdominant (γ and δ) epitopes that were shown to be immunogenic by IFN-γ ELISpot. **c** Plot of $S_b$ and $S_i$ of predicted HLA-A*02:01 epitopes for the SpCas9 protein. Red dots represent the immunodominant and subdominant epitopes. **d** IFN-γ ELISpot assay of T cell reactivity of 12 healthy donors (the two non- HLA-A*02:01 are shown as open circles) to 38 predicted epitopes grouped in 10 pools, CEF (positive control peptide pool), and DMSO (negative control). Peptides α and δ were in pool 5 while β and γ were in pool 3. **e** IFN-γ ELISpot reactivity of healthy donor T cells (*n* = 12) to epitopes across the different domains of the Cas9 protein. Donors 1–10 were HLA-A*02:01, while 11 and 12 were not. Peptides α and δ overlap in 5 amino acid residues. Data represent mean ± SD. EBNA-1, Epstein-Barr virus nuclear antigen-1; $S_b$, normalized binding score; $S_i$, normalized immunogenicity score. Statistical analysis was performed post hoc and results are exploratory. Source data are available in the Source Data file

epitopes identified by IFN-γ ELISpot were within the top 5 most immunogenic epitopes predicted by our immunogenicity model[32]. Their ranking as predicted by the consensus method hosted on the IEDB server using default settings was 14, 5, 18, and 4, respectively. Sequence similarity of peptides α and β to amino acid sequences in known proteins was investigated using Protein BLAST and the IEDB epitope database[34]. This was done to investigate whether there is any chance that the T cell immune

response that we are detecting in healthy individuals could be due to previous exposure to another protein of similar sequence. A peptide was considered 'similar' to α or β if no more than 2 of 9 amino acid residues (that are not the second or ninth) were not matching (78% similarity). None of these two peptides resembled known epitopes in the IEDB database, but similarity to other Cas9 orthologs and other bacterial proteins was detected (Supplementary Tables 2 and 3). Epitope β has sequence

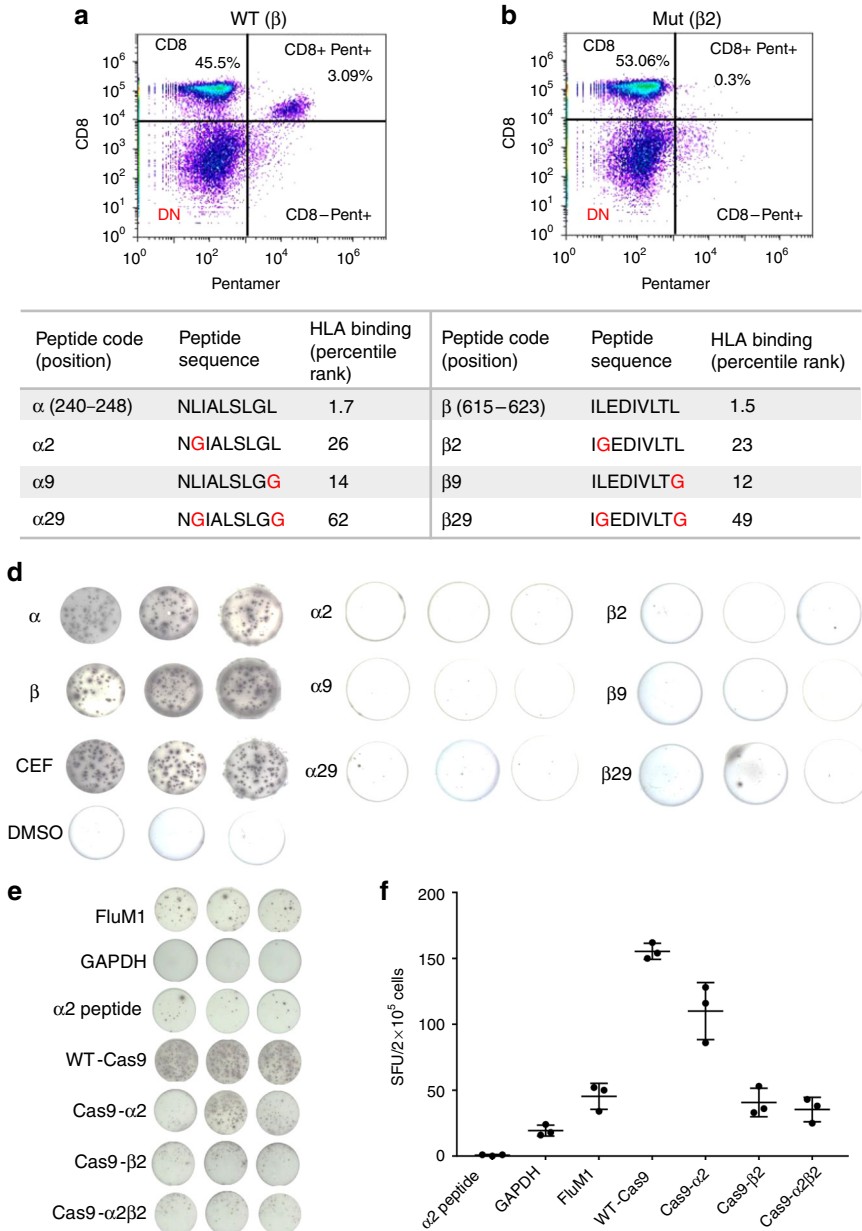

| Peptide code (position) | Peptide sequence | HLA binding (percentile rank) | Peptide code (position) | Peptide sequence | HLA binding (percentile rank) |
|---|---|---|---|---|---|
| α (240–248) | NLIALSLGL | 1.7 | β (615−623) | ILEDIVLTL | 1.5 |
| α2 | NGIALSLGL | 26 | β2 | IGEDIVLTL | 23 |
| α9 | NLIALSLGG | 14 | β9 | ILEDIVLTG | 12 |
| α29 | NGIALSLGG | 62 | β29 | IGEDIVLTG | 49 |

**Fig. 2** SpCas9 immunodominant epitope-specific CD8+T cell recognition is abolished after anchor residue mutation. **a** Epitope β-specific CD8+T cell response detected using β-specific pentamer in PBMCs stimulated with peptide β-pulsed antigen presenting cells. **b** The percentage of CD8+ pentamerβ+T cells was reduced to 0.3% when healthy donor B cell APCs were pulsed with the mutated peptide β2. **c** Positions, sequences, and IEDB HLA binding percentile rank of epitopes α and β before and after mutation of the anchor (2nd and/or 9th) residues. $S_b$, normalized binding score; $S_i$, normalized immunogenicity score. **d** Representative IFN-γ ELISpot assay in triplicate wells comparing T cell reactivity to wild type or mutated epitopes α and β. These results are representative of 12 donors and two independent replicates (data from all 12 donors are shown in Supplementary Fig. 1). **e, f** IFN-γ ELISpot comparing T cell reactivity to APCs expressing WT or modified Cas9 proteins. APCs expressing FluM1 were used as a positive control. APCs expressing GAPDH or spiked with peptide α2 were used as negative controls. Data represent mean ± SEM of 5 replicates (right). Statistical analysis was performed post hoc and results are exploratory. Source data are available in the Source Data file

similarity to a peptide derived from the *Neisseria meningitidis* peptide chain release factor 2 protein (ILEDIVLTL versus ILE**G**IVLTL). Antigen-specific T cells were expanded for 18 days in vitro by coculturing healthy donor PBMCs with peptide β-pulsed autologous antigen presenting cells (APCs). Cas9-specific CD8+T cell responses were assessed by flow cytometry. CD8+T cells specific for the HLA-A*0201/β pentamer were detected after stimulation (3.09%; Fig. 2a).

**Mutated Cas9 proteins have lower immune recognition.** We next hypothesized that mutation of the MHC-binding anchor

residues of the identified immunogenic epitopes would abolish specific T cell recognition (Fig. 2a). The epitope anchor residues (2nd and 9th) are not only necessary for peptide binding to the MHC groove, but are also crucial for recognition by the T cell receptor[32]. The percentage of CD8+β pentamer+T cells decreased to 0.3% when APCs were pulsed with the mutated peptide (β2; Fig. 2b) compared with 3.09% with the wild type peptide (β; Fig. 2a). We then examined the reactivity of healthy donor T cells to modified peptides α or β with mutations in residues 2, 9, or both (sequences are shown in Fig. 2c) using IFN-γ ELISpot assay. The epitope-specific T cell reactivity was

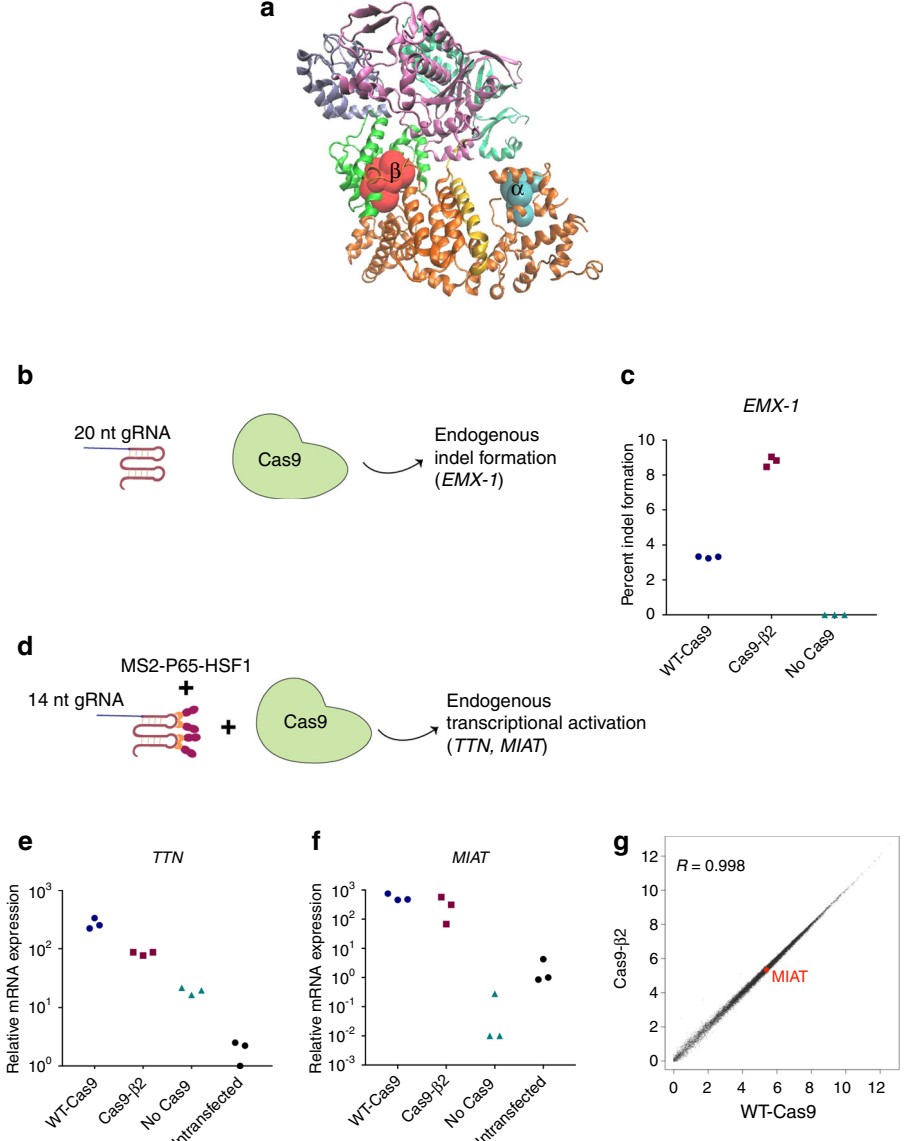

**Fig. 3** Mutated SpCas9 protein (Cas9-β2) retains its function and specificity. **a** 3D structure of the SpCas9 protein, showing the location of the identified immunodominant epitopes α and β. **b** Schematic of the experiment assessing mutagenesis capacity of Cas9-β2. Cells were transfected with either WT-Cas9, Cas9-β2, or an empty plasmid as well as 20 nt gRNA targeting *EMX-1* locus. 72 h after transfection, percent cleavage was assessed by DNA extraction and illumina sequencing. **c** Percentage of indel formation in *EMX-1* locus. Data represent mean ± SD of three individual transfections. **d** Schematic of the experiment assessing gRNA binding, DNA targeting and transcriptional modulation with Cas9-β2. Cells were transfected with either WT-Cas9, Cas9-β2, or an empty plasmid as well as 14nt gRNA targeting *TTN* or *MIAT* in the presence of MS2-P65-HSF1 (transcriptional modulation). 72 h after transfection, mRNA was assessed by qRT-PCR. **e, f** Shown is the mRNA level relative to an untransfected control experiment ($n = 3$ independent technical replicaes). **g** Mean expression levels of 24,078 protein-coding and non-coding RNA genes for WT-Cas9 and Cas9-β2 (each in duplicate) are shown. For visualization purposes, the values were transformed to a $\log_2(\text{CPM}+1)$ scale. *MIAT*, the gRNA target gene, is highlighted in red, and R denotes Pearson correlation coefficient between two groups. Source data are available in the Source Data file

markedly reduced with the mutant peptides (Supplementary Fig. 1; representative ELISPOT well images are shown in Fig. 2d). The average reduction for the responsive HLA-A*02:01 donors was 25-fold from α to α29 ($n = 7$; $p < 0.03$) and 30-fold from β to β29 ($n = 8$; $p < 0.03$; Benjamini-Hochberg; Supplementary Fig. 1; Supplementary Table 4). The predicted binding affinity to MHC class II was also decreased for α2 and β2 epitopes, although the experimental significance of this alteration is unknown.

We then generated modified Cas9 constructs by mutating the second residue of peptide α (L241G; Cas9-α2), peptide β (L616G; Cas9-β2), or both (Cas9-α2β2). To measure the effect of mutating the anchor residue of the immunogenic epitopes on T cell

recognition of the Cas9 protein, we transiently transfected healthy donor B cell APCs with mRNA encoding wild type Cas9 (WT-Cas9), Cas9-α2, Cas9-β2, or Cas9-α2β2. The T cell response measured by IFN-γ ELISpot after coculturing of Cas9 transfected APCs with autologous PBMCs was significantly decreased for the modified Cas9 proteins (Fig. 2e, f). Introduction of the β2 mutation was the most effective in reducing T cell immunogenicity (5.5-fold, $p < 0.0001$, two-tailed *t*-test). This mutation in the REC1 domain (Figs. 1e and 3a) is not located in any of the two regions that are absolutely essential for DNA cleavage, the repeat-interacting (97–150) and the anti-repeat-interacting (312–409) regions[33]. These results demonstrate that mutating the anchor

amino acid of a highly immunogenic epitope can influence the overall immunogenicity of Cas9. Thus, engineering Cas9 variants with reduced immunogenicity potential can be used in conjunction with other strategies for safer CRISPR therapies and even possibly reduce the dosage of systemic immunosuppression needed for patients.

We therefore tested the function of Cas9-β2 in comparison with WT-Cas9 in the context of DNA cleavage and transcriptional modulation. To examine the nuclease activity of Cas9-β2 and compare with WT-Cas9, we targeted Cas9-β2 or WT-Cas9 to an endogenous locus (EMX-1) and measured percent indel formation (Fig. 3b, c). Our data demonstrate that Cas9-β2 retains nuclease capacity in the locus we studied as well as on a synthetic promoter (Fig. 3c, Supplementary Fig. 2A, B).

Next, we determined whether Cas9-β2 can successfully recognize and bind its target DNA leading to transcriptional modulation. We first tested this in the context of enhanced transgene expression from a synthetic CRISPR responsive promoter in HEK293 cells using 14nt gRNAs and aptamer-mediated recruitment of transcriptional modulators similar to what we had shown before (Supplementary Fig. 2C, D). Having shown successful transgene activation, we then investigated whether this variant retains such capacity within the chromosomal contexts of endogenous genes. We transfected the cells with plasmids encoding Cas9-β2 or WT-Cas9 and 14nt gRNAs against two different endogenous genes (TTN and MIAT). qRT-PCR analysis showed that this variant successfully led to target gene expression (Fig. 3d–f). To further characterize Cas9-β2 specificity, we performed genome-wide RNA sequencing after targeting Cas9-β2 or WT-Cas9 to the MIAT locus for transcriptional activation. The results demonstrated no significant increase in undesired off-target activity by Cas9-β2 as compared to WT-Cas9 (Fig. 3g).

To show the extensibility of our approach, we tested the function of Cas9-α2, that has a mutation located in the REC2 domain (Figs. 1e and 3a). Cas9-α2 also demonstrated DNA cleavage and transcriptional modulation functionality comparable with WT-Cas9 (Supplementary Fig. 3A–E). This is consistent with a previous study which showed that Cas9 with a deleted REC2 domain retains its nuclease activity[33]. When T cells were stimulated with APCs spiked with peptide α2, the percentage of CD8+CD137+T cells (a marker of T cell activation[35]) was decreased by 2.3-fold as compared to WT peptide α stimulation (Supplementary Fig. 3F).

**Immune responses to non-HLA-A*02:01 Cas9 epitopes**. We next predicted T cell epitopes derived from SpCas9 for non-HLA-A*02:01 alleles using the IEDB analysis tool (Supplementary Data 1). We selected and synthesized 5–6 epitopes (sequences shown in Supplementary Table 5 and highlighted in Supplementary Data 1) for each of 7 common alleles (A*01:01, A*03:01, A*11:01, A*24:02, B*08:01, B*44:01, B*55:01) and used them to stimulate PBMCs derived from 6 healthy donors with the corresponding HLA alleles. Of the peptides and alleles screened, we detected peptide-specific T cell immune responses (in more than one donor) against peptide 25 in two HLA-A*24:02 donors ($p < 0.05$; results for HLA-A*24:02 are shown as an example in Fig. 4a).

**Immune responses to MHC Class II Cas9 epitopes**. We also predicted MHC class II binding epitopes for the SpCas9 protein to HLA-DRB1 (10 alleles), HLA-DQ (5 alleles), and HLA-DP (8 alleles) using the IEDB analysis tool (Supplementary Data 2). For MHC class II, epitope α is predicted to be a top binder to HLA-DRB1*01:02 and epitope β a top binder to HLA-DPA1*01:03 and DPB1*02:01 (Supplementary Data 2). We selected and

synthesized SpCas9 long peptides that include epitopes in the top 2% of predicted MHC class II binders (sequences and alleles shown in Supplementary Table 6 and highlighted in Supplementary Data 2) and measured peptide-specific CD4+ T cell immunity using IFN-γ secretion ELISpot assays with CD8-depleted PBMCs derived from three healthy individuals. We detected limited CD4+ immune responses against the peptides tested (Fig. 4b). This prompted us to evaluate the CD4+ T cell immune response against the whole Cas9 protein compared with a positive control protein. We stimulated healthy donor CD8-depleted PBMCs with recombinant SpCas9 or EBNA proteins for 10 days and detected a modest response (less than twofold of that of unstimulated cells; Fig. 4c). We then sought to investigate whether the immune response that we detected against peptide β was primarily derived by CD4+ or CD8+ T lymphocytes. We stimulated MACS sorted CD4+ or CD8+ T cells isolated from PBMCs from 3 donors with peptide β and detected a primarily CD8+ response in all 3 donors (Fig. 4d).

**Discussion**

The detection of pre-existing B cell and T cell immunity to the most widely used nuclease ortholog of the CRISPR-Cas9 tool in a significant proportion of healthy humans confirms previous studies in mice[9,10] and a recent study in humans[27] and sheds light on the need for more studies of the immunological risks of this system. A recent study reported SpCas9-specific T-cell-mediated killing of autologous CRISPR-treated APCs in vitro[28]. However, the consequences of Cas9-specific immunity in vivo remain to be seen. The CD8+T cell immunity we observed is likely memory responses, as they are observed without ex vivo stimulation. However, following 18 days of T cell stimulation by peptides α or β, expansion of naïve T cells is not precluded. This suggests that even in the absence of a pre-existing immune response, the expression of Cas9 in naïve individuals may trigger a T cell response that could prevent subsequent administration. This could be avoided by switching to Cas9 orthologs from other bacterial species, but can be difficult given the epitope conservation across Cas9 proteins from multiple Streptococcus species and resemblance to sequences from other bacterial proteins such as the common pathogen N. meningitidis, that asymptomatically colonizes the nasopharynx in 10% of the population[36]. Therefore, selective deimmunization (immunosilencing) of Cas9 can represent an attractive alternative, particularly in patients where high-dose systemic immunosuppression is contraindicated, such as in patients with chronic infectious diseases. This strategy can be important in particular when longer term expression of Cas9 will be desired.

Using IFN-γ ELISpot, we detected a modest CD4+ immune response against recombinant SpCas9 protein and no response to any of the class II SpCas9 peptides that we used. This is consistent with a recent study that found the majority of the CD4+ response against SpCas9 to be restricted to the Treg compartment with minimum IFN-γ and TNF-α secretion[28]. While a more comprehensive epitope mapping study is needed, we did not experimentally detect immunodominant epitopes among the top binders of the non-HLA-A*02:01 alleles. We show here that silencing one epitope for an HLA-A*02:01 donor (who was also positive for A*11:01, B*39:01, and B*46:01) was sufficient to significantly reduce Cas9 immunogenicity. However, whether other mutations need to be introduced to the protein for complete silencing in HLA-A*02:01 negative individuals needs further investigation. Conventional methods of deimmunizing non-human therapeutic proteins rely on trial-and-error mutagenesis or machine learning and often includes deletion of whole regions of the protein[37–41]. Here, as a general principle, we show that

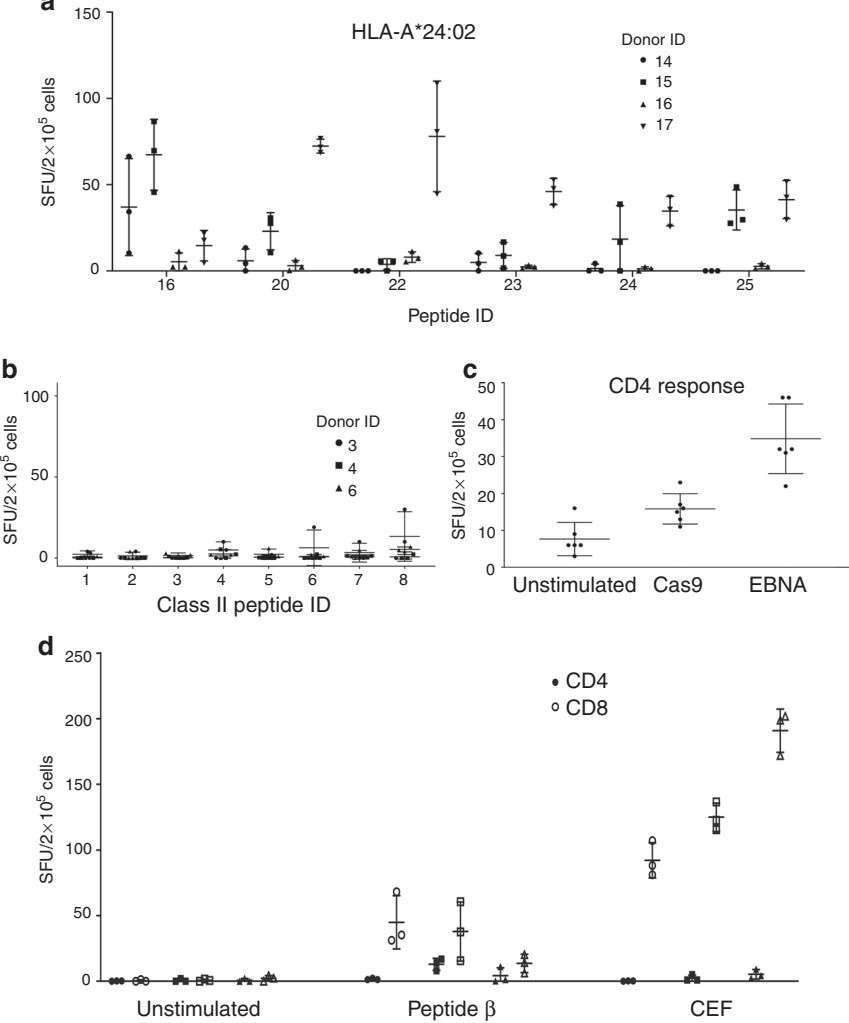

**Fig. 4** Immune responses to non-HLA-A*02:01 and class II epitopes of SpCas9. **a** Representative IFN-γ ELISpot reactivity of healthy donor T cells to non-HLA-A*02:01 SpCas9 class I epitopes (HLA-A*024:02 is shown as an example). **b** IFN-γ ELISpot reactivity of healthy donor MACS-sorted CD4+ T cells to SpCas9 long peptides that include epitopes in the top 2% of predicted MHC class II binders. **c** IFN-γ ELISpot reactivity of healthy donor CD8-depleted PBMCs stimulated with recombinant SpCas9 or EBNA proteins for 10 days. **d** IFN-γ ELISpot reactivity of MACS sorted CD4+ (black dots) or CD8+ (open dots) T cells isolated from PBMCs from three healthy donors unstimulated or stimulated with peptide β or CEF. Data represent mean ± SD. Source data are available in the Source Data file

alteration of one of the anchor residues of an immunodominant epitope abolished specific T cell recognition. However, HLA allotype diversity and the existence of numerous epitopes in the large Cas9 protein may complicate the process of complete deimmunization. The overall impact of removal of select immunodominant epitopes remains to be seen; both reduction[42] and enhancement[43] of the immunogenicity of subdominant epitopes have been reported with similar approaches for other proteins.

The top binding T cell epitopes within Cas9 that are most promiscuous for common HLA class I and class II alleles have been recently predicted in silico using IEDB[44]. However, this is the first study that experimentally validates predicted immunodominant epitopes. None of the HLA-A*02:01 epitopes that we report overlap with the peptides previously predicted[44]. This is not surprising since our prediction model is optimized for HLA-A*02:01. Thus, improved algorithms are needed to predict epitopes that hold up in experimental validation, as we show here. The use of CRISPR-Cas9 in humans may eventually necessitate creating HLA type-specific Cas9 variants, particularly for applications that require long-term Cas9 expression.

Non-specific localized immune suppressive approaches, such as those used by tumor cells and some viruses may complement these strategies for complete deimmunization. Antigen presentation can be blocked by viral proteins interfering with antigen presentation (VIPRs), such as the adenoviral E319K or US2 and US11 from the human cytomegalovirus[45] or molecules that inhibit proteasomal antigen processing such as the Epstein-Barr virus Gly-Ala repeat[46]. Deimmunized Cas9 may be useful in reduction of the dosage of other immunomodulatory measures needed to be co-administered in patients, thus facilitating therapeutic CRISPR applications as we develop better understanding of the immunological consequences of this system.

## Methods

**Detection of Cas9-specific serum antibodies**. Healthy control sera ($n = 143$) used in this study, and previously[47], are a subset of a molecular epidemiology study of head and neck cancer at the MD Anderson Cancer Center, collected between January 2006 and September 2008. Samples were collected using a standardized sample collection protocol and stored at -80 °C until use. All relevant ethical regulations for work with human participants were followed and written informed consent was obtained from all participants under the Arizona State University institutional review board approval. Participants were ethnically diverse. Sample

size was determined based on prior experience and similar experiments in the literature. There was no group allocation and the investigators were blinded to participants' information. *S. pyogenes* lysate was prepared by sonication of bacterial pellets from overnight cultures of *S. pyogenes* ATCC 19615 in the presence of 1 pill of cOmplete Protease Inhibitor (Sigma-Aldrich) after 3 cycles of freezing and thawing. Serum antibody detection was performed using ELISA. 96-well plates were coated with 20 μg/mL of recombinant *S. pyogenes* Cas9 nuclease (Takara Bio USA, Mountain View, CA), recombinant Epstein-Barr virus nuclear antigen 1 (EBNA; Advanced Biotechnologies Inc., Eldersburg, MD), human hemoglobin (Sigma-Aldrich) or *S. pyogenes* lysate. Sera were diluted 1:50 in 10% *E. coli* lysate prepared in 5% milk-PBST (0.2% tween)[48], incubated with shaking for 2 h at room temperature, and added to the specified wells in duplicate. Horseradish peroxidase (HRP) anti-human IgG Abs (Jackson ImmunoResearch Laboratories, West Grove, PA) were added at 1:10,000, and detected using Supersignal ELISA Femto Chemiluminescent substrate (Thermo Fisher Scientific, Waltham, MA). Luminescence was detected as relative light units (RLU) on a Glomax 96 Microplate Luminometer (Promega, Madison, WI) at 425 nm. The cutoff value was defined as any reactivity higher than the top 99% of RLU values for human hemoglobin (Fig. 1a, dotted line).

**Cas9 candidate T cell epitope prediction.** We predicted MHC class I restricted 9-mer and 10-mer candidate epitopes derived from the Cas9 protein (Uniprot - Q99ZW2) for HLA A*02:01. The protein reference sequence was entered into 5 different prediction algorithms; 3 MHC-binding: IEDB-consensus binding[49], NetMHCpan binding[50], Syfpeithi[51], and 2 antigen-processing algorithms: IEDB-consensus processing, ANN processing[52]. The individual scores from each of the prediction algorithms were then normalized within the pool of predicted peptides after exclusion of poor binders, and the average normalized binding scores were used to re-rank the candidate peptides. The top 38 candidate peptides (Supplementary Table 1) were selected for experimental testing. The IEDB consensus MHC-binding prediction algorithm (http://www.iedb.org/) was applied to obtain a list of high binding Cas9 peptides, each of which was assigned a normalized binding score ($S_b$). The immunogenicity score ($S_i$) was calculated for each peptide based on its amino acid hydrophobicity (ANN-Hydro)[32]. MHC class II epitopes were predicted using the recommended setting on IEDB. We used the Visual Molecular Dynamics software[53] (http://www.ks.uiuc.edu/Research/vmd/) to generate the Cas9 protein structure (PDB ID: 4CMP, Fig. 3a).

**Ex vivo stimulation and epitope mapping of Cas9 by ELISpot.** All peripheral blood mononuclear cells (PBMCs) were obtained from healthy individuals with written informed consent under the Arizona State University institutional review board and all relevant ethical regulations for work with human participants were followed. Participants were ethnically diverse. Sample size was determined based on prior experience and similar experiments in the literature. There was no group allocation and the investigators were blinded to participants' information. If PBMCs from a given donor were not reactive to the positive control, the donor was excluded from the study. PBMCs were isolated from fresh heparinized blood by Ficoll–Hypaque (GE Healthcare, UK) density gradient centrifugation and stimulated. Briefly, predicted Cas9 peptides with $S_b < 0.148$ ($n = 38$) were synthesized (>80% purity) by Proimmune, UK. Each peptide was reconstituted at 1 mg/mL in sterile PBS and pools were created by mixing 3–4 candidate peptides. Sterile multiscreen ELISpot plates (Merck Millipore, Billerica, MA, USA) were coated overnight with 5 μg/well of anti-IFN-γ capture antibody (clone D1K, # 3420-3-250, Mabtech, USA) diluted in sterile PBS. Frozen PBMCs were thawed rapidly and recombinant human IL-2 (20 U/mL, R&D Systems) was added. They were then stimulated in triplicates with 10 μg/mL Cas9 peptide pools (or individual peptides), pre-mixed CEF pool as a positive control (ProImmune, UK), or DMSO as a negative control in the anti-IFN-γ-coated ELISpot plates, (Merck Millipore, Billerica, MA, USA) and incubated in a 37 °C, 5% $CO_2$ incubator for 48 h. Plates were washed three times for 5 min each with ELISpot buffer (PBS+0.5% FBS) and incubated with 1 μg/mL anti-IFN-γ secondary detection antibody (clone 7-B6-1, # 3420-6-250, Mabtech, USA) for 2 h at room temperature, washed and incubated with 1 μg/mL Streptavidin ALP conjugate for 1 h at room temperature. The wells were washed again with ELISpot buffer and spots were developed by incubating for 8–10 min with detection buffer (33 μL NBT, 16.5 μL BCIP, in 100 mM Tris-HCl pH 9, 1 mM $MgCl_2$, 150 mM NaCl). Plates were left to dry for 2 days and spots were read using the AID ELISpot reader (Autoimmun Diagnostika GmbH, Germany). The average number of spot forming units for each triplicate was calculated for each test peptide or peptide pool and subtracted from the background signal.

**Autologous APC generation from healthy individual PBMCs.** Autologous CD40L-activated B cell APCs were generated from healthy donors by incubating whole PBMCs with irradiated (32 Gy) K562-cell line expressing human CD40L (KCD40L) at a ratio of 4:1 (800,000 PBMCs to 200,000 irradiated KCD40Ls) in each well. The cells were maintained in B cell media (BCM) consisting of IMDM (Gibco, USA), 10% heat-inactivated human serum (Gemini Bio Products, CA, USA), and Antibiotic-Antimycotic (Anti-Anti, Gibco, USA). BCM was supplemented with 10 ng/mL recombinant human IL-4 (R&D Systems, MN, USA), 2 μg/mL Cyclosporin A (Sigma-Aldrich, CA, USA), and insulin transferrin supplement (ITES, Lonza, MD, USA). APCs were re-stimulated with fresh irradiated KCD40Ls on days 5 and 10, after washing with PBS and expanding into a whole 24-well plate. After two weeks, APC purity was assessed by CD19+ CD86+ expressing cells using flow cytometry, and were used for T cell stimulation after >90% purity. APCs were either restimulated up to 4 weeks or cryopreserved for re-expansion as necessary.

**T cell stimulation by autologous APCs.** Antigen-specific T cells were detected by stimulating healthy donor B cell APCs by either peptide pulsing of specific Cas9 epitopes, or by transfecting with mRNA encoding the whole WT or modified Cas9 proteins. Peptide pulsing of APCs was done under BCM 5% human serum, with recombinant IL-4. Transfection of APCs was done with primary P3 buffer in a Lonza 4D Nucleofector and program EO117 (Lonza, MD, USA) and incubated in BCM-10% human serum and IL-4. Twenty-four hrs later, on day 1, APCs were washed and incubated with thawed whole PBMCs at a ratio of 1:2 (200,000 APCs: 400,000 PBMCs) in a 24-well plate in BCM supplemented with 20 U/mL recombinant human IL-2 (R&D Systems, MN, USA) and 5 ng/mL IL-7 (R&D Systems, MN, USA). On day 5, partial media exchange was performed by replacing half the well with fresh BCM and IL-2. On day 10, fresh APCs were peptide pulsed in a new 24-well plate. On day 11, expanded T cells were restimulated with peptide-pulsed APCs similar to day 1. T cells were used for T cell assays or immunophenotyped after day 18.

**Flow cytometry staining for T cells.** Cells were washed once in MACS buffer (containing PBS, 1% BSA, 0.5 mM EDTA), centrifuged at 550 g for 5 min and re-suspended in 200 μL MACS buffer. Cells were stained in 100 μL of staining buffer containing anti-CD137, conjugated with phycoerythrin (PE, clone 4B4-1; BD Biosciences, USA), anti-CD8-PC5 (clone B9.11; Beckman Coulter 1:100), anti-CD4 (clone SK3; BioLegend, 1:200), anti-CD14 (clone 63D3; BioLegend, 1:200), and anti-CD19 (clone HIB19; BioLegend,1:200), all conjugated to Fluorescein isothiocyanate (FITC) for exclusion gates, for 30 min on ice. Samples were covered and incubated for 30 min on ice, washed twice in PBS, and resuspended in 1 mL PBS prior to analysis.

**Pentamer staining for T cell immunophenotyping.** The following HLA-A*02:01 PE-conjugated Cas9 pentamers were obtained from ProImmune: F2A-D-CUS-A*02:01-ILEDIVLTL-Pentamer and 007-Influenza A MP 58-66-GILGFVFTL-Pentamer. T cells were washed twice in MACS buffer with 5% human serum and centrifuged at 550 g for 5 min each time. They were then re-suspended in 100 μL staining buffer (MACS buffer, with 5% human serum and 1 mM Dasatanib (ThermoFisher Scientific, MA, USA). Each of the pentamers was added to resuspended T cells, stimulated with the respective peptide or APCs at a concentration of 1:100. Samples were incubated at room temperature for 30 min in the dark, then washed twice in MACS buffer. Cells were stained in 100 μL MACS buffer with anti-CD8-PC5, anti-CD4-FITC, anti-CD14-FITC, and anti-CD19-FITC for exclusion gates, Samples were then washed twice with PBS and analyzed by flow cytometry. For flow cytometric analysis, all samples were acquired with Attune flow cytometer (ThermoFisher Scientific, MA, USA) and analyzed using the Attune software. Gates for expression of different markers and pentamers were determined based on flow minus one (FMO) samples for each color after doublet discrimination (Supplementary Fig. 4). Percentages from each of the gated populations were used for the analysis.

Modified Cas9 plasmids: human codon-optimized *Streptococcus pyogenes* Cas9 sequence was amplified from pSpCas9 (pX330; Addgene plasmid ID: 42230), using forward and reverse primers and inserted within gateway entry vectors using golden gate reaction. Desired mutations were designed within gBlocks (Integrated DNA Technologies). The gBlocks and amplicons were then cloned into gateway entry vectors using golden gate reaction. All the primer sequences are listed in Supplementary Table 7 and all the gBlocks sequences are listed in Supplementary Note 1. Next, the Cas9 vectors and CAG promoter cassettes were cloned into an appropriate gateway destination vector via LR reaction (Invitrogen).

U6-sgRNA-MS2 plasmids: these plasmids were constructed by inserting either 14 bp or 20 bp spacers of gRNAs into sgRNA (MS2) cloning backbone (Addgene plasmid ID: 61424) at BbsI site. All the gRNA sequences are listed in Supplementary Table 8.

**Cell culture for endogenous target mutation and activation.** HEK293FT cell line was purchased from ATCC and maintained in Dulbecco's modified Eagle's medium (DMEM - Life Technologies) containing 10% fetal bovine serum (FBS - Life Technologies), 2 mM glutamine, 1 mM sodium pyruvate (Life Technologies), and 1% penicillin-streptomycin (Life Technologies) in incubators at 37 °C and 5% $CO_2$. Polyethylenimine (PEI) was used to transfect HEK293FT cells seeded into 24-well plates. Transfection complexes were prepared according to manufacturer's instructions (Polysciences).

**Fluorescent reporter assay for quantifying Cas9 function.** For the experiment assessing Cas9 cleavage capacity at a synthetic promoter, HEK293FT cells were co-transfected with 200 ng gRNA, 200 ng Cas9 constructs, 50 ng reporter plasmid, and 25 ng enhanced blue florescent protein (EBFP) expressing plasmid as the transfection control. For the experiment assessing Cas9 transcriptional activation

capacity at a synthetic promoter, HEK293FT cells were co-transfected with 50 ng gRNA, 70 ng Cas9 constructs, 100 ng MS2-P65-HSF1-GFP (Addgene plasmid ID: 61423), 200 ng reporter plasmid. Fluorescent reporter experiments were performed 48 h after transfection. Nonviable cells were excluded from the analysis using 7-AAD (7-amino-actinomycin D) conjugated with PerCP. Next, we selected cells expressing EBFP >$2 \times 10^2$ A.U. or GFP >$2 \times 10^2$ A.U. (transfection markers) in the cleavage and activation experiments, respectively to exclude untransfected cells (Supplementary Fig. 4). Flow cytometry was performed using a FACSCelesta flow cytometer (Becton Dickson) with HTS. Flow cytometry data were analyzed using FlowJo software. Untransfected controls were included in each experiment. Experiments underwent initial validation in duplicate and then were repeated in triplicate for the final manuscript.

**Quantitative RT-PCR analysis.** HEK293FT cells were co-transfected with 10 ng gRNA, 200 ng Cas9 constructs, 100 ng MS2-P65-HSF1 (Addgene plasmid ID: 61423) and 25 ng EBFP plasmid as the transfection control. Cells were lysed, and RNA was extracted using RNeasy Plus mini kit (Qiagen) 72 h post transfection, followed by cDNA synthesis using the High-Capacity RNA-to-cDNA Kit (Thermo Fisher). qRT-PCR was performed using SYBR Green PCR Master Mix (Thermo Fisher). All analyses were normalized to 18S rRNA ($\Delta Ct$) and fold changes were calculated against untransfected controls ($2^{-\Delta\Delta Ct}$). Primer sequences for qPCR are listed in Supplementary Table 7.

**Endogenous indel analysis.** HEK293FT cells were co-transfected with 200 ng of Cas9 plasmids, 10 ng gRNA coding cassette, and 25 ng EBFP plasmid as the transfection control. 72 h later, transfected cells were dissociated and spun down at 200 g for 5 min at room temperature. Genomic DNA was extracted using 50 µl of QuickExtract DNA extraction solution (Epicentre) according to the manufacturer's instructions. Genomic DNA was amplified by PCR using primers flanking the targeted region. Illumina Tru-Seq library was created by ligating partial adapters and a unique barcode to the DNA samples. Next, a small number of PCR cycles was performed to complete the partial adapters. Equal amounts of each sample were then pooled and sequenced on Illumina Tru-Seq platform with $2 \times 150$ run parameters, which yielded approximately 80,000 reads per sample. Sequencing was performed using a $2 \times 150$ paired-end (PE) configuration by CCIB DNA Core Facility at Massachusetts General Hospital (Cambridge, MA, USA). The reads were aligned to the target gene reference in *Mus musculus* genome using Geneious software, 9-1-5. To detect the indels (insertions and deletions of nucleic acid sequence at the site of double-strand break), each mutation was evaluated carefully in order to exclude the ones that are caused by sequencing error or any off-target mutation. The variant frequencies (percentage to total) assigned to each read containing indels were summed up. i.e. indel percentage = total number of indel containing reads/ total number of reads. The minimum number of analyzed reads per sample was 70,000.

**RNA sequencing for quantifying activator specificity.** HEK293FT cells were co-transfected with 10 ng gRNA targeting *MIAT* locus, 200 ng Cas9 constructs, 100 ng MS2-P65-HSF1 (Addgene plasmid ID: 61423), and 25 ng transfection control. Total RNA was extracted 72 h post transfection using RNeasy Plus mini kit (Qiagen) and sent to UCLA TCGB core on dry ice. Ribosomal RNA depletion, and single read library preparation were performed at UCLA core followed by RNA sequencing using NextSeq500. Coverage was 14 million reads per sample. FASTQ files with single-ended 75 bp reads were then aligned to the human GRCh38 reference genome sequence (Ensembl release 90) with STAR[54], and uniquely-mapped read counts (an average of 14.8 million reads per sample) were obtained with Cufflink[55]. The read counts for each sample were then normalized for the library size to CPM (counts per million reads) with edgeR[56]. Custom R scripts were then used to generate plots.

**Reporting summary.** Further information on experimental design is available in the Nature Research Reporting Summary linked to this article.

## Data availability

The datasets generated and/or analyzed during the current study have been submitted to the Gene Expression Omnibus (GEO) repository and are accessible through GEO series accession number GSE125522 . Source data for this manuscript are available in the Source Data file. All other relevant data are available from the authors upon reasonable request.

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

## Acknowledgements

We thank Dr. Erich Sturgis for providing the serum samples, Dr. Diego Chowell for support with the computational prediction model, and Padhmavathy Yuvaraj for technical assistance. We thank all the other members of the Anderson, Kiani, and Ebrahimkhani labs for their assistance and insightful discussions. This work was supported by ASU institutional funds, the startup fund by the School of Biological and Health Systems Engineering of ASU, National Institute of Biomedical Imaging and Bioengineering R01 grant (1R01EB024562-01A1), and DARPA Young Faculty Award (D16AP00047) to S.K. We thank the Center for Computational and Integrative Biology (CCIB) DNA core facility at Massachusetts General Hospital and Genomics and Bioinformatics (TCGB) core at UCLA for the DNA and RNA sequencing services.

## Author contributions

S.R.F. and R.E. designed experiments, performed experiments, and analyzed data. F.M. generated Cas9 variant constructs, performed Cas9 functional analysis experiments, and analyzed data. S. Krishna generated predicted Cas9 epitopes and designed and assisted with the T cell experiments. F.M. and S. Krishna contributed equally. J.G.P. analyzed RNA seq data. M.R.E. helped with the design of experiments and interpretation of data. S.K. and K.S.A. supervised this study. R.E. took the lead in writing the manuscript with input from S.R.F., S.K. and K.S.A. and support of all the other authors.

## Additional information

**Competing interests:** The Arizona Board of Regents on behalf of Arizona State University has submitted a patent application (application number: PCT/US18/29937, filed 27 April 2018), with authors S.R.F., R.E., F.M., S.K., M.R.E., S.K. and K.S.A. The patent application was published on 24 Jan 2019 and it is awaiting national stage entry. It includes a methods and compositions for reducing an undesirable T cell immune response in human patients before or during gene therapy using the CRISPR/Cas9-based genetic modulation. Also, it includes DNA sequences of cas9-α2, -β2, -α2β2, and sequences of guide RNAs and primers. The remaining authors declare no competing interests.

