## [Peer Review File · Nature Communications]

Reviewers' Comments:

Reviewer #1:

Remarks to the Author:

This is a revision of the manuscript entitled, "Multifunctional CRISPR/Cas9 with engineered immunosilenced human T cell epitopes" from Ferdosi et al.

Overall, the authors did a very good job of responding to my comments, adding MHC class II information. I have no other changes to the manuscript, and I am okay with publication.

Reviewer #2:

Remarks to the Author:

Major points

The authors now include predicted CD4 T cell epitopes in the manuscript. They do not, however, show the respective experimental data. This is a bit disappointing, as they introduce the story of immunity against Cas9 with very nice data about antibodies recognizing the protein in serum of human donors. As this B cell response against a monovalent antigen must be CD4 T cell supported, a T cell assay, best with cells from the same donors and using predicted peptides, would certainly enhance the story significantly. Pre-existing helper cell activity may also be critical for mounting a CD8 T cell driven immune response upon viral delivery of Cas9. Further, the authors did not connect the story lines "pre-existing immunity against Cas9" and "gene therapy induced immunity against Cas9" sufficiently.

The authors restricted their analysis and protein re-design to only one MHC allele. Can they estimate, for how many alleles redesign may be possible? For how many not (because the epitope lies in an essential part of the protein)? Such an analysis would greatly enhance the value of the manuscript as it would indicate whether the approach is applicable rather generally and not only to the lucky few patients with the right MHC.

Comment 3.9: Why was the flow cytometry performed without dead cell staining/exclusion?

Response 3.9: Thank you for your comment. In the experiments involving testing Cas9 functionality by transfecting synthetic gene circuits, we co-transfect the cells with an independent fluorescent protein. During Flow cytometry analysis, we gated the cells based on FSC/SSC properties, gating out cellular debris, then gate the cells that express this independent marker. The independent fluorescent marker serves as a proxy for cells that are alive and transfected. We then look at the expression of the output signal. This is a standard practice in the field for experiments involving synthetic gene circuits. In other experiments, cells were over 95% alive by trypan blue exclusion before being run on flow cytometry. Hence, just double discrimination was sufficient.

NEW COMMENT: Bad experimental design by others should not be used to justify own design. 95% alive means 5% dead (and possibly sticky). There is no good reason to not perform live/dead staining, unless one is running out of colors and is extremely skilled and careful about controls. I do not see this here. Especially when using MHC reagents care needs to be taken.

Figure 3E: 3 independent transfections (on the same day) or three independent experiments (different days)?

Figure 2D: Data should be quantified and displayed, as done for Figure 2 E.

Minor points

The authors should have a more careful approach about the future. Whether RNPs or viral delivery will dominate the field remains to be seen (and may depend on the application). They should adjust their writing accordingly.

Where possible, box plots with dots should be used instead of bar graphs. E.g. Fig 2, Sup. Fig. 1
Comment 3.7: Were experiments F, H, I truly performed 3x independently?

Response 3.7: The experiment was repeated 2 times independently with different conditions. And the one that is submitted is one experiment with n=3 biological replicates.

NEW COMMENT: All data should be shown, indicate technical versus biological replicates by color or filling.

Comment 3.11: Where possible, box plots with dots should be used instead of bar graphs.

Response 3.11: Thank you. This has been adjusted based on your comment.

NEW COMMENT: NOT DONE

Comment 3.15: Supp Fig 2 Were the T-tests corrected for multiple testing where necessary?

Response 3.15: Thank you for the comment. We now applied multiple T-test correction using the Benjamini-Hochberg method.

NEW COMMENT: No mention of the BHM could be found in the manuscript. The BHM works on the level of the FDR – were the uncorrected p values shown (and significance according to BHM indicated)?

Comment 3.25: They should remove the PDL1 idea at the end of discussion. Not relevant to the story and dangerous for the patient.

Response 3.25: Thank you for raising this point. What we suggest here is transient inducible coexpression of PD-L1 activating gRNAs inside cells that will express Cas9 to protect them against cytolysis by T cells. This is one of several approaches that can be used to mitigate Cas9 immunogenicity. We now expand more on these strategies in the manuscript discussion, last paragraph.

NEW COMMENT: The authors want to sell a story of low immunogenicity Cas9 protein. Why they insist on bringing about a completely different immunomodulatory approach without evidence in the paper, remains mysterious. They should consider, that if they want to follow down that line in the lab and have it published before (in this paper), they will not be able to patent, rendering the concept commercially useless.

The flow cytometry reporting guidelines of Nature should be followed:

<https://www.nature.com/authors/policies/FlowCytometry.pdf>

NEW COMMENT: NOT DONE. E.G. axis labels don't conform.

Sequences and sequencing data should be uploaded to respective databases and access numbers indicated in the manuscript.

NEW COMMENT: Was it done?

Page 3 "A peptide was considered "similar" to". Where did this cut-off for similarity come from. Is there underlying evidence that can be cited?

Reviewer #3:

Remarks to the Author:

I could not evaluate authors' answers to 3 of the 4 concerns that I raised in the previous review because of the following reasons.

Antibody titer experiment in Fig 1A needs a known protein antigen positive control, which would be recognized by these donors, and a true negative control, instead of "To establish cut-off values, a RLU ratio > (the mean + 2 standard deviations) of all samples with signal below the mean RLU (horizontal black lines) was designated positive". These controls are needed to demonstrate the pre-existing antibodies to either S. Pyogenes lysate or Cas9 could be at a physiological significant

level.

I think the data on Cas9 specific T cells in healthy donors would be significantly strengthened if authors assess the frequency in CD4 and CD8 T cells separately. I also noticed that the Y axis label has been changed from $SFU \times 10^6$ to $SFU/2 \times 10^5$ without any other change on Y scale nor on data point position on this figure. I recognize the previous label, $SFU \times 10^6$, was a typo. It should be $SFU/10^6$. So, the new Y axis label makes the existing T cell frequency 5 times higher.

Similarly, Supplementary Figure 1 (original Supplementary Figure 2) was change on the Y axis label but not Y axis scale. However, frequency bars for some of the panel A donors remained the same, while others have drastic changes. Panel B, frequency bars among some donors seemed to be switched.

Authors need to explain these changes that are important to the data interpretation as one of my concern raised in the previous review was whether pre-existing T cell immunity was an issue to begin with and if so, whether Cas9 mutation really can reduce or eliminate it.

Minor: CEF used in the Figure caption needs to be spelled out.

Response to Referees:

Below we provide a point-by-point response including corresponding descriptions of revisions to the manuscript. For Reviewer #2, for clarity, we have included both the prior comments and, in bold, the new comments and responses to the reviewer.

Reviewer #1 (Remarks to the Author):

This is a revision of the manuscript entitled, "Multifunctional CRISPR/Cas9 with engineered immunosilenced human T cell epitopes" from Ferdosi et al.

Overall, the authors did a very good job of responding to my comments, adding MHC class II information. I have no other changes to the manuscript, and I am okay with publication.

Reviewer #2 (Remarks to the Author):

Major points

NEW COMMENT 2.1: The authors now include predicted CD4 T cell epitopes in the manuscript. They do not, however, show the respective experimental data. This is a bit disappointing, as they introduce the story of immunity against Cas9 with very nice data about antibodies recognizing the protein in serum of human donors. As this B cell response against a monovalent antigen must be CD4 T cell supported, a T cell assay, best with cells from the same donors and using predicted peptides, would certainly enhance the story significantly. Pre-existing helper cell activity may also be critical for mounting a CD8 T cell driven immune response upon viral delivery of Cas9.

NEW RESPONSE 2.1: Thank you for this comment. We have selected and synthesized long peptides that include epitopes in the top 2% of predicted MHC class II binders (sequences are shown in Supplementary Table 8 and highlighted in Supplementary Table 6). We measured peptide-specific CD4⁺ T cell immunity using IFN- γ secretion ELISpot assays with CD8-depleted PBMCs derived from 3 healthy individuals. We did not detect significant CD4⁺ immune responses against any of the peptides tested (Fig. 4B). This prompted us to evaluate the CD4⁺ T cell immune response against the whole

Cas9 protein compared with a positive control protein. We stimulated healthy donor CD8-depleted PBMCs with recombinant SpCas9 or EBNA proteins for 10 days and detected a modest response (less than twofold of that of unstimulated cells; Fig. 4C). We then sought to investigate whether the immune response that we detected against peptide β was primarily derived by CD4⁺ or CD8⁺ T lymphocytes. We stimulated MACS-sorted CD4⁺ or CD8⁺ T cells isolated from PBMCs from 3 donors with peptide β and detected a primarily CD8⁺ response in all 3 donors (Fig. 4D). These results are consistent with a recent study, published since our manuscript submission, that found the majority of the CD4⁺ response against SpCas9 to be restricted to the Treg compartment with minimum IFN- γ and TNF- α secretion (Wagner et al., Nat Med 2018).

NEW COMMENT 2.2: Further, the authors did not connect the story lines “pre-existing immunity against Cas9” and “gene therapy induced immunity against Cas9” sufficiently.

NEW RESPONSE 2.2: This has now been clarified in the first paragraph of the discussion, page 5.

NEW COMMENT 2.3: The authors restricted their analysis and protein re-design to only one MHC allele. Can they estimate, for how many alleles redesign may be possible? For how many not (because the epitope lies in an essential part of the protein)? Such an analysis would greatly enhance the value of the manuscript as it would indicate whether the approach is applicable rather generally and not only to the lucky few patients with the right MHC.

NEW RESPONSE 2.3: Thank you for this comment. We now include predictions for T cell epitopes derived from SpCas9 for non-HLA-A*02:01 alleles using the IEDB analysis tool (Supplementary Table 5). We selected and synthesized 5-6 epitopes (sequences shown in Supplementary Table 7 and highlighted in Supplementary Table 5) for each of 7 common alleles (A*01:01, A*03:01, A*11:01, A*24:02, B*08:01, B*44:01, B*55:01) and used them to stimulate PBMCs derived from 6 healthy donors with the corresponding HLA alleles. Of all the alleles and peptides tested, we detected immune reactivity in more than one donor to only one epitope for one allele. While non-HLA-A*:02:01 alleles need a more comprehensive study, we could not experimentally detect immunodominant epitopes among the top binders of the alleles we tested as predicted by the IEDB method. We show in our study (figure 2) that silencing one epitope for an HLA-A*02:01 donor (who was also positive for A*11:01, B*39:01, and B*46:01) was enough to significantly reduce the immunogenicity. However, whether other mutations need to be introduced to the protein for complete silencing in HLA-A*02:01 negative individuals needs further investigation.

Comment 3.9: Why was the flow cytometry performed without dead cell staining/exclusion?

Response 3.9: Thank you for your comment. In the experiments involving testing Cas9

functionality by transfecting synthetic gene circuits, we co-transfect the cells with an independent fluorescent protein. During Flow cytometry analysis, we gated the cells based on FSC/SSC properties, gating out cellular debris, then gate the cells that express this independent marker. The independent fluorescent marker serves as a proxy for cells that are alive and transfected. We then look at the expression of the output signal. This is a standard practice in the field for experiments involving synthetic gene circuits. In other experiments, viability >95% by trypan blue exclusion was required prior to flow cytometry.

NEW COMMENT 2.4: Bad experimental design by others should not be used to justify own design. 95% alive means 5% dead (and possibly sticky). There is no good reason to not perform live/dead staining, unless one is running out of colors and is extremely skilled and careful about controls. I do not see this here. Especially when using MHC reagents care needs to be taken.

NEW RESPONSE 2.4: Thank you for your comment. The experiments involving testing Cas9 functionality have now been repeated based on your comment. We now include live-dead staining (7-AAD). We gate the cells based on both a transfection marker and the 7-AAD marker in flow cytometry and measure the output of the circuit in the presence of WT-Cas9 or Cas9 variants. In figure 2A and B, we had used fluorescence minus one (FMO) controls as demonstrated in Supplementary Figure 4B.

NEW COMMENT 2.5: Figure 3E: 3 independent transfections (on the same day) or three independent experiments (different days)?

NEW RESPONSE 2.5: The data presented in this graph shows three individual transfections performed on the same day.

NEW COMMENT 2.6: Figure 2D: Data should be quantified and displayed, as done for Figure 2 E.

NEW RESPONSE 2.6: Thank you for your comment. As explained in Figure 2 legend, Figure 2D is a representative example of one of the 12 donors and two independent replicates quantified and displayed in Figure 2E. Now we have added the word 'representative' to figure 2D legend to clarify this.

Minor points

NEW COMMENT 2.7: The authors should have a more careful approach about the future. Whether RNPs or viral delivery will dominate the field remains to be seen (and may depend on the application). They should adjust their writing accordingly.

NEW RESPONSE 2.7: Thank you for your comment. We adjusted the manuscript to reflect this caution. See results section, paragraph 2.

Comment 3.7: Were experiments F, H, I truly performed 3x independently?

RESPONSE 3.7: The data presented in Fig. 3 shows three individual transfections per condition.

NEW COMMENT 2.8: All data should be shown, indicate technical versus biological replicates by color or filling.

NEW RESPONSE 2.8: All the dots in the experiments involving Cas9 functional tests represent one biological replicate (individual transfection).

Comment 3.11: Where possible, box plots with dots should be used instead of bar graphs.

Response 3.11: Thank you. This has been adjusted based on your comment.

NEW COMMENT 2.9: NOT DONE (box plots with dots should be used instead of bar graphs. E.g. Fig 2, Sup. Fig. 1)

NEW RESPONSE 2.9: This has now been done for Figure 2 and Supplementary Figure 1.

Comment 3.15: Supp Fig 2 Were the T-tests corrected for multiple testing where necessary?

Response 3.15: Thank you for the comment. We now applied multiple T-test correction using the Benjamini-Hochberg method.

NEW COMMENT 2.10: No mention of the BHM could be found in the manuscript. The BHM works on the level of the FDR – were the uncorrected p values shown (and significance according to BHM indicated)?

NEW RESPONSE 2.10: Thank you for this comment. We now show the uncorrected and corrected p values for donors with a significant reduction in T cell response after mutating epitope α or β by the Benjamini-Hochberg method in Supplementary Table 4.

Comment 3.25: They should remove the PDL1 idea at the end of discussion. Not relevant to the story and dangerous for the patient.

Response 3.25: Thank you for raising this point. What we suggest here is transient inducible coexpression of PD-L1 activating gRNAs inside cells that will express Cas9 to protect them against cytolysis by T cells. This is one of several approaches that can be used to mitigate Cas9 immunogenicity. We now expand more on these strategies in the manuscript discussion, last paragraph.

NEW COMMENT 2.11: The authors want to sell a story of low immunogenicity Cas9 protein. Why they insist on bringing about a completely different immunomodulatory approach without evidence in the paper, remains mysterious. They should consider, that if they want to follow down that line in the lab and have it published before (in this paper), they will not be able to patent, rendering the concept commercially useless.

NEW RESPONSE 2.11: This has now been deleted from the last paragraph of the discussion page 6.

The flow cytometry reporting guidelines of Nature should be

followed: <https://www.nature.com/authors/policies/FlowCytometry.pdf>

NEW COMMENT 2.12: NOT DONE. E.G. axis labels don't conform.

NEW RESPONSE 2.12. This has been adjusted.

Sequences and sequencing data should be uploaded to respective databases and access numbers indicated in the manuscript.

NEW COMMENT 2.13: Was it done?

NEW RESPONSE 2.13. These will be submitted after acceptance.

NEW COMMENT 2.14: Page 3 "A peptide was considered "similar" to". Where did this cut-off for similarity come from. Is there underlying evidence that can be cited?

NEW RESPONSE 2.14: Because the anchor residues (2nd and 9th) are important for peptide binding to the MHC groove and for recognition by the TCR, we excluded peptides that were similar to α or β but that were different in their anchor residues. We believe that these are less likely to trigger a cross-reactive T cell immune response. Since we were comparing sequence similarity of 9-mers, we considered a difference of 3 of 9 amino acids (66.7% similarity) to also be too different. To our knowledge, this is the first study to examine the effect of mutation of one anchor amino acid residue on whole protein immunogenicity and function.

Reviewer #3 (Remarks to the Author):

I could not evaluate authors' answers to 3 of the 4 concerns that I raised in the previous review because of the following reasons.

Comment 3.1. Antibody titer experiment in Fig 1A needs a known protein antigen positive control, which would be recognized by these donors, and a true negative control, instead of "To establish cut-off values, a RLU ratio > (the mean + 2 standard deviations) of all samples with signal below the mean RLU (horizontal black lines) was designated positive". These controls are needed to demonstrate the pre-existing antibodies to either *S. Pyogenes* lysate or Cas9 could be at a physiological significant level.

Response 3.1. Thank you for this comment. We repeated this experiment for the same serum samples for Cas9 with EBNA-1 as a positive control protein and human hemoglobin as a negative control protein. We now calculate the cutoff as any reactivity higher than the top 99% of RLU values for human hemoglobin. This has now been changed in the results page 2, figure 1A and its legend page 11, and the methods page 13. For our repetition, we used a recombinant SpCas9 protein from a different commercial source (Takara Bio USA, Mountain View, CA).

Comment 3.2. I think the data on Cas9 specific T cells in healthy donors would be significantly strengthened if authors assess the frequency in CD4 and CD8 T cells separately.

Response 3.2. Thank you for this comment. We now added an experiment to address this. We stimulated MACS sorted CD4⁺ or CD8⁺ T cells isolated from PBMCs from 3 donors with peptide β and detected a primarily CD8⁺ response in all 3 donors (**Fig. 4D**). Also, please see response 2.1.

Comment 3.3. I also noticed that the Y axis label has been changed from SFU $\times 10^6$ to SFU/2 $\times 10^5$ without any other change on Y scale nor on data point position on this figure. I recognize the previous label, SFU $\times 10^6$, was a typo. It should be SFU/10⁶. So, the new Y axis label makes the existing T cell frequency 5 times higher. Similarly, Supplementary Figure 1 (original Supplementary Figure 2) was change on the Y axis label but not Y axis scale. However, frequency bars for some of the panel A donors remained the same, while others have drastic changes. Panel B, frequency bars among some donors seemed to be switched.

Response 3.3. Thank you for this comment. We now realized that our explanation in response 3.3 in the previous letter to the editor was not detailed enough. The Y axis scales of figures 1D and 1E were originally SFU/2 $\times 10^5$ and SFU/10⁶, respectively. However, the label on figure 1D was mislabeled SFU $\times 10^6$. In our revision, we corrected figure 1D label and changed the scale of figure 1E to SFU/2 $\times 10^5$ to be the same as 1D for easier comparison.

We did change the numbering for donors 1-12 as we mentioned in Response 3.4 in the previous letter to the editor. The non-HLA-A*02:01 donors became donors 11 and 12 in figure 1E and Supplementary Figure 1 (both panels A and B) for easier comparison of the HLA-A*02:01 donors (i.e. donors 5, 10, 11, and 12 in our first submission became donors 12, 11, 9, and 10 in the revision, respectively). Based on reviewer's comment 3.4, we excluded donor 9 (in the old version) because it was not HLA typed and we included data from a new HLA-typed donor, which is now donor 5. There was a typographical error in the value reported for peptide α (blue bar) for donor 12 (now donor 10), which is now corrected.

Authors need to explain these changes that are important to the data interpretation as one of my concern raised in the previous review was whether pre-existing T cell immunity was an issue to begin with and if so, whether Cas9 mutation really can reduce or eliminate it.

Comment 3.4. Minor: CEF used in the Figure caption needs to be spelled out.

Response 3.4. This has now been added to figure 1 legend page 11.

Reviewers' Comments:

Reviewer #2:

Remarks to the Author:

The manuscript has improved significantly. Only minor changes need to be undertaken.

- a) The figure legends should indicate clearly whether the data comes from technical or biological replicates. Doing an identical transfection 3 times on the same day is a technical replicate. This information is very important for the reader.
- b) No prospective power analysis was done. It should again in the figure legends stated clearly that all statistics was done posthoc and the results are exploratory.
- c) Fig. 2D. The authors do not want to bring the data together in a graph but display only representative data. This should still be changed. In figure 2E the authors did exactly what should also be done for 2D.
- d) Sample allocation, randomization and blinding have to mentioned in M&M section (if done or not done), not only on the separate form (later not accessible to the reader).
- e) The data presentation of the FACS form is not filled.

Thorsten Buch

Reviewer #3:

Remarks to the Author:

Revision addressed my concerns. However, CEF (positive control) should be listed as CEF (positive control peptide pool) in Figure 1 caption.

January 18, 2019

Dear Dr. Cloney,

We would like to thank the reviewers for their positive and informative review of this manuscript. Detailed answers to the points raised are given below.

Reviewer #2 (Remarks to the Author):

The manuscript has improved significantly. Only minor changes need to be undertaken.

a) The figure legends should indicate clearly whether the data comes from technical or biological replicates. Doing an identical transfection 3 times on the same day is a technical replicate. This information is very important for the reader.

Thank you for this comment. This has been adjusted in the figure legends.

b) No prospective power analysis was done. It should again in the figure legends stated clearly that all statistics was done posthoc and the results are exploratory.

Thank you for this comment. This has been added to the legends of figures 1 and 2 and Supplementary Figure 1.

c) Fig. 2D. The authors do not want to bring the data together in a graph but display only representative data. This should still be changed. In figure 2E the authors did exactly what should also be done for 2D.

Thank you for this comment. All the data from the 12 donors are shown in Supplementary Figure 1, and due to limited space, we only show representative well images in the main figure (2D). We now clarify this in the results (under "Mutated Cas9 proteins have lower immune recognition and maintain their function and specificity, page 3) and in Fig. 2 legend page 15 (highlighted text).

d) Sample allocation, randomization and blinding have to mentioned in M&M section (if done or not done), not only on the separate form (later not accessible to the reader).

This has now been added to the methods pages 6 – 9 (highlighted text).

e) The data presentation of the FACS form is not filled.

This has now been filled out.

Reviewer #3 (Remarks to the Author):

Revision addressed my concerns. However, CEF (positive control) should be listed as CEF (positive control peptide pool) in Figure 1 caption.

Thank you for this comment. This has been added to Figure 1 legend page 15.

Reviewers' Comments:

Reviewer #2:

Remarks to the Author:

All my concerns were addressed.